# Differentiable Combinatorial Losses through Generalized Gradients of Linear Programs

## Abstract

Combinatorial problems with linear objective function play a central role in many computer science applications, and efficient algorithms for solving them are well known. However, the solutions to these problems are not differentiable with respect to the parameters specifying the problem instance – for example, shortest distance between two nodes in a graph is not a differentiable function of graph edge weights. Recently, attempts to integrate combinatorial and, more broadly, convex optimization solvers into gradient-trained models resulted in several approaches for differentiating over the solution vector to the optimization problem. However, in many cases, the interest is in differentiating over only the objective value, not the solution vector, and using existing approaches introduces unnecessary overhead. Here, we show how to perform gradient descent directly over the objective value of the solution to combinatorial problems. We demonstrate advantage of the approach in examples involving sequence-to-sequence modeling using differentiable encoder-decoder architecture with softmax or Gumbel-softmax, and in weakly supervised learning involving a convolutional, residual feed-forward network for image classification.

## 1 Introduction

Combinatorial optimization problems, such as shortest path in a weighted directed graph, minimum spanning tree in a weighted undirected graph, or optimal assignment of tasks to workers, play a central role in many computer science applications. We have highly refined, efficient algorithms for solving these fundamental problems (Cormen et al., 2009; Schrijver, 2003). However, while we can easily find, for example, the minimal spanning tree in a graph, the total weight of the tree as function of graph edge weights is not differentiable. This problem hinders using solutions to combinatorial problems as criteria in training models that rely on differentiability of the objective function with respect to the model parameters.

Losses that are defined by objective value of some feasible solution to a combinatorial problem, not the optimal one, have been recently proposed for image segmentation using deep models (Zheng et al., 2015; Lin et al., 2016). These focus on a problem where some pixels in the image have segmentation labels, and the goal is to train a convolutional network that predicts segmentation labels for all pixels. For pixels with labels, a classification loss can be used. For the remaining pixels, a criterion based on a combinatorial problem – for example the maximum flow / minimal cut problem in a regular, lattice graph connecting all pixels (Boykov et al., 2001) or derived, higher-level super-pixels (Lin et al., 2016) – is often used as a loss, in an iterative process of improving discrete segmentation labels (Zheng et al., 2015; Marin et al., 2019). In this approach, the instance of the combinatorial problem is either fixed, or depends only on the input to the network; for example, similarity of neighboring pixel colors defines edge weights. The output of the neural network gives rise to a feasible, but rarely optimal, solution to that fixed instance a combinatorial problem, and its quality is used as a loss. For example, pixel labeling proposed by the network is interpreted as a cut in a pre-defined graph connecting then pixels. Training the network should result in improved cuts, but no attempt to use a solver to find an optimal cut is made.

Here, we are considering a different setup, in which each new output of the neural network gives rise to a new instance of a combinatorial problem. A combinatorial algorithm is then used to find the optimal solution to the problem defined by the output, and the value of the objective function of

the optimal solution is used as a loss. After each gradient update, the network will produce a new combinatorial problem instance, even for the same input sample. Iteratively, the network is expected to learn to produce combinatorial problem instances that have low optimal objective function value. For example, in sequence-to-sequence modeling, the network will output a new sentence that is supposed to closely match the desired sentence, leading to a new optimal sequence alignment problem to be solved. Initially, the optimal alignment will be poor, but as the network improves and the quality of the output sentences get higher, the optimal alignment scores will be lower.

Recently, progress in integrating combinatorial problems into differentiable models have been made by modifying combinatorial algorithms to use only differentiable elements (Tschiatschek et al., 2018; Mensch & Blondel, 2018; Chang et al., 2019), for example smoothed max instead of max in dynamic programming. Another approach involves executing two runs of a non-differentiable, black-box combinatorial algorithm and uses the two solutions to define a differentiable interpolation (Vlastelica Pogančić et al., 2020; Rolínek et al., 2020). Finally, differentiable linear programming and quadratic programming layers, which can be used to model many combinatorial problems, have been proposed recently (Amos & Kolter, 2017; Agrawal et al., 2019; Wilder et al., 2019; Ferber et al., 2019).

The approaches above allow for differentiating through optimal solution vectors. In many cases, we are interested only in the optimal objective value, not the solution vector, and the approaches above introduce unnecessary overhead. We propose an approach for gradient-descent based training of a network $f(x; \beta)$ for supervised learning problems involving samples $(x, y)$ with the objective criterion involving a loss term of the form $L(\beta) = h(\text{OptSolutionObjectiveValue}(\Pi(F(x; \beta), y))$, where $h : \mathbb{R} \to \mathbb{R}$ is some differentiable function, and $\Pi$ is a combinatorial solver for a problem instance defined by the output of the $\beta$-parameterized network $F$ for feature vector $x$ and by the true label $y$. We show that a broad class of combinatorial problems can be integrated into models trained using variants of gradient descent. Specifically, we show that for an efficiently solvable combinatorial problem that can be efficiently expressed as an integer linear program, generalized gradients of the problem's objective value with respect to real-valued parameters defining the problem exist and can be efficiently computed from a single run of a black-box combinatorial algorithm. Using the above result, we show how generalized gradients of combinatorial problems can provide sentence-level loss for text summarization using differentiable encoder-decoder models that involve softmax or Gumbel softmax (Jang et al., 2016), and a multi-element loss for training classification models when only weakly supervised, bagged training data is available.

## 2 Differentiable Combinatorial Losses

### 2.1 Background on Generalized Gradients

A function $f : \mathcal{X} \to \mathbb{R}$ defined over a convex, bounded open set $\mathcal{X} \in \mathbb{R}^p$ is Lipschitz continuous on an open set $B \in \mathcal{X}$ if there is a finite $K \in \mathbb{R}$ such that $\forall x, y \in B \ |f(x) - f(y)| \leq K||x - y||$. A function is locally Lipschitz-continuous if for every point $x_0$ in its domain, there is a neighborhood $B_0$, an open ball centered at $x_0$, on which the function is Lipschitz-continuous. For such functions, a generalized gradient can be defined.

**Definition 1.** (Clarke, 1975) *Let* $f : \mathcal{X} \to \mathbb{R}$ *be Lipschitz-continuous in the neighborhood of* $x \in \mathcal{X}$. *Then, the* Clarke subdifferential $\partial f(x)$ *of* $f$ *at* $x$ *is defined as*

$$\partial f(x) = \text{conv} \left\{ \lim_{x_k \to x} \nabla f(x_k) \right\},$$

*where the limit is over all convergent sequences involving those* $x_k$ *for which gradient exists, and* conv *denotes convex hull, that is, the smallest polyhedron that contains all vectors from a given set. Each element of the set* $\partial f(x)$ *is called a* generalized gradient *of* $f$ *at* $x$.

The Rademacher theorem (see e.g. (Evans, 1992)) states that for any locally Lipschitz-continuous function the gradient exists almost everywhere; convergent sequences can be found.

In optimization algorithms, generalized gradients can be used in the same way as subgradients (Redding & Downs, 1992), that is, nondifferentiability may affect convergence in certain cases.

## 2.2 Gradient Descent over Combinatorial Optimization

Many combinatorial problems have linear objective function and can be intuitively expressed as integer linear programs (ILP), that is, linear programs with additional constraint that the solution vector involves only integers. Any ILP can be reduced to a linear program. Consider an ILP

$$z^* = ILP(c, A', b') := \min_u \ c^T u \ \text{ s.t. } A'u = b', \ u \geq 0, \ u \in \mathbb{Z}^p,$$

with an optimal solution vector $u^*$ and optimal objective value $z^*$. Then, there exists a corresponding linear program $LP(c, A, b)$

$$z^* = LP(c, A, b) := \min_u \ c^T u \ \text{ s.t. } Au = b, \ u \geq 0,$$

called *ideal formulation* (Wolsey, 1989), for which $u^*$ is also an optimal solution vector, with the same objective value $z^*$. For a feasible, bounded $p$-dimensional integer program, we can view the pair $(A', b')$ as a convex polyhedron $\mathcal{A}'$, the set of all feasible solutions. Then, the pair $(A, b)$ in the ideal formulation LP is defined as the set of constraints specifying the feasible set $\mathcal{A} = \text{conv} \{\mathcal{A}' \cap \mathbb{Z}^p\}$. Convex hull of a subset of a convex set $\mathcal{A}'$ cannot extend beyond $\mathcal{A}'$, thus, $\mathcal{A}$ is convex, contains all integer solutions from $\mathcal{A}'$, and no other integer solutions. The number of linear constraints in the ideal formulation may be exponential in $p$, and/or in $m$, the number of the original constraints in $\mathcal{A}'$. Thus, the existence of the ideal formulation LP for an ILP may not have practical utility for solving the ILP.

For a combinatorial problem and its corresponding ILP, we use the ideal formulation of the ILP as a conceptual tool to define generalized gradient of the objective value of the optimal solution to the combinatorial problem with respect to the parameters defining the combinatorial problem. Specifically, our approach first uses a single run of an efficient, black-box combinatorial algorithm to produce the optimal solution vector and the associated objective value. Then, the combinatorial problem is conceptually viewed as an instance of an ILP. A possibly exponentially large linear program (LP) equivalent to the ILP is then used, without actually being spelled out or solved, to derive generalized gradients based on the solution vector returned by the combinatorial algorithm.

First, we introduce several notions of efficiency of transforming a combinatorial problem into a linear integer program that will be convenient in defining the generalized gradients of combinatorial problems.

**Definition 2.** *Let $P(w)$ be a combinatorial problem that is parameterized by a continuous vector $w \in \mathcal{W} \subseteq \mathbb{R}^n$, where $\mathcal{W}$ is simply connected and $n$ is the problem size, and let $k \in \mathbb{Z}$ be a constant that may depend on the problem type but not on its size. Then, a combinatorial problem is*

- primal-dual $\partial$-efficient *if it can be phrased as an integer linear program involving $n$ variables, with $kn$ constraints in an LP formulation equivalent to the ILP, and the parameters $(A, b, c)$ of the LP formulation depend on $w$ through (sub)differentiable functions, $c = c(w), A = A(w), b = b(w)$.*

- primal $\partial$-efficient *if it can be phrased as an integer linear program involving $n$ variables, the parameters $w$ of the problem influence the cost vector $c$ through a (sub)differentiable function $c = c(w)$, and do not influence the constraints $A, b$.*

- dual $\partial$-efficient *if it can be phrased as an integer linear program in which the number of constraints in the equivalent LP formulation is $kn$, the parameters $w$ of the problem influence $b$ through a (sub)differentiable function $b = b(w)$, and do no influence the constraint matrix $A$ nor the cost vector $c$.*

The class of $\partial$-efficient problems includes polynomially solvable combinatorial problems with objective function that is linear in terms of problem parameters. Typically, the functions $c = c(w)$, $b = b(w)$ and $A = A(w)$ are either identity mapping or are constant; for example, in the LP for maximum network flow, the cost vector $c$ is composed directly of edge capacities, and $A$ an $b$ are constant for a given flow network topology, and do not depend on capacities.

For any polynomially solvable combinatorial problem, we can construct a $\text{poly}(n)$-sized Boolean circuit for the algorithm solving it. For each $\text{poly}(n)$-sized circuit, there is a linear program with $\text{poly}(n)$ variables and constraints that gives the same solution (see (Dasgupta et al., 2008), Chap.

7). For example, for MST in a graph with $V$ vertices and $E$ edges, the Martin's ILP formulation (Martin, 1991) has only $\text{poly}(V + E)$ constraints, but it is an extended formulation that involves $VE$ additional variables on top of the typical $E$ variables used in the standard ILP formulations for MST. Thus, we cannot use it to construct an ILP formulation that would make MST primal-dual $\partial$-efficient. Alternatively, there is an ILP for MST with one binary variable per edge, and the weight of the edge only influences the cost vector $c$, but to prohibit cycles in the solution there is a constraint for each cycle in the graph, thus the number of constraints is not $\text{poly}(n)$ for arbitrary graphs. These constraints are specified fully by the topology of the graph, not by the edge weights, so $w$ does not influence $A$ nor $b$, meeting the conditions for primal $\partial$-efficiency. The MST example shows that there are problems that are primal $\partial$-efficient and not primal-dual $\partial$-efficient.

Some polynomially solvable combinatorial problems are not $\partial$-efficient in any of the above sense. For example, fixed-rank combinatorial problems with interaction costs (Lendl et al., 2019) can be phrased succinctly as a bilinear program, but lead to prohibitively large linear programs both in terms of the number of variables and the number of constraints.

For $\partial$-efficient problems, we can efficiently obtain generalized gradients of the objective value.

**Theorem 1.** *Consider a combinatorial problem $P(w)$ of size $n$, a parameter vector $w$ from the interior of the parameter domain $\mathcal{W}$, and an algorithm $\Pi(w)$ for solving it in time $\text{poly}(n)$. Let $z^*$ be the optimal objective value returned by $\Pi$. Then,*

- *if $P$ is primal $\partial$-efficient, then the generalized gradients $\partial z^*(w)$ exist, and can be efficiently computed from $U^*$, the set of primal solution of the ideal formulation of integer program corresponding to $P$;*

- *if $P$ is dual $\partial$-efficient, then the generalized gradients of $\partial z^*(w)$ exist, and can be efficiently computed from $V^*$, the set of all dual solution to the ideal formulation of the integer program corresponding to $P$;*

- *if $P$ is primal-dual $\partial$-efficient, then the generalized gradients of $A$ over $w$ exist, and can be efficiently computed from $U^*$ and $V^*$, as defined above.*

*Proof.* A series of results (Gal, 1975; Freund, 1985; De Wolf & Smeers, 2000) shows that if the optimal objective value $z^* = LP(c, A, b)$ for a linear program is finite at $(c, A, b)$ and in some neighborhood of $(c, A, b)$, then generalized gradients of $z^*$ with respect to $c$, $b$, and $A$ exist and are

$$\partial z^*(c) = U^*, \ \ \partial z^*(b) = V^*, \ \ \partial z^*(A) = \left\{ -vu^T : (u, v) \in V^* \times U^* \right\}.$$

We build on these results to obtain generalized gradients of the linear program corresponding to the combinatorial problem. For the first case in the theorem, definition 2 states that in the linear program corresponding to $P$, only the cost vector $c$ depends on $w$, through a (sub)differentiable function $c = c(w)$. Since $w$ is in the interior of the parameter domain $\mathcal{W}$, the objective value is finite over some neighborhood of $w$. Then,

$$\partial z^*(w) = \partial z^*(c) \frac{\partial c}{\partial w} = \frac{\partial c}{\partial w} U^*,$$

where the generalized gradient $z^*(c)$ exists and is equal to $U^*$.
For the second case, the ideal formulation LP exists. Then, from definition 2 we have that

$$\partial z^*(w) = \partial z^*(b) \frac{\partial b}{\partial w} = \frac{\partial b}{\partial w} V^*.$$

The third case is a direct extension of the first two cases. □

Theorem 1 indicates that black-box combinatorial algorithms can be used to expand the range of transformations that can be efficiently utilized in neural networks. One immediate area of application is using them to specify a loss function. Consider a network $F(x; \beta)$ parameterized by a vector of tunable parameters $\beta$. The network transforms a batch of input samples $x$ into a batch of outputs $\chi = F(x; \beta)$. Then, in the broadest primal-dual $\partial$-efficient case, $\chi$ is used, possibly with the true classes $y$, to formulate parameters $(c, A, b) = g(\chi, y)$ of a linear program corresponding to the combinatorial problem, through some (sub)differentiable function $g$. For

---

**Algorithm 1** Minimization of a combinatorial loss

---

**Input:** batch $x \subset \mathcal{X}$, $y \subset \mathcal{Y}$, network $F(x; \beta)$, functions $g, h$, combinatorial algorithm $\Pi$
**Output:** Loss and its generalized gradient, $L(\beta), \partial L(\beta)$

1: **procedure** COMBLOSSMIN($x, y, \beta, F, g, h, \Pi$)
2:     forward pass $\chi = F(x; \beta)$
3:     forward pass $(c, A, b) = g(\chi, y)$
4:     run combinatorial solver to find optimal objective value $z^* = \Pi(c, A, b)$ and optimal primal
          and/or dual solution vectors $u^*, v^*$
5:     forward pass $L(\beta) = h(z^*)$
6:     backward pass through $h$: $\partial L / \partial z^*$
7:     backward pass through $\Pi$: $\partial z^*(c) = u^*$, $\partial z^*(b) = v^*$, $\partial z^*(A) = -v^* u^{*T}$
8:     backward pass through $g$ and $F$
9:     $\partial L(\beta) = \frac{\partial L}{\partial z} \left( u^* \frac{\partial c}{\partial \beta} - v^* u^{*T} \frac{\partial A}{\partial \beta} + v^* \frac{\partial b}{\partial \beta} \right)$
10:    **return** $L(\beta), \partial L(\beta)$
11: **end procedure**

---

a given $\beta$ and given batch samples $(x, y)$, we can then define loss as a function of the optimal objective value of the linear program corresponding to the combinatorial problem resulting from $g(F(x; \beta), y)$, $L(\beta) = h(z^*(c, A, b))$. This approach, summarized in Algorithm 1, allows us to obtain the generalized gradient of the loss with respect to $\beta$ as long as functions $g$ and $h$ are differentiable. For clarity, in Algorithm 1, we did not consider functions $h$ depending not just on $z$ but also on $x$ or $y$, but the extension is straightforward.

## 3 EXAMPLE USE CASES AND EXPERIMENTAL VALIDATION

### 3.1 DIFFERENTIATING OVER BIPARTITE MATCHING FOR WEAKLY-SUPERVISED LEARNING

To illustrate gradient descent over a combinatorial loss, we first focus on a simple image recognition problem. Consider a photo of a group of people with a caption listing each of the persons in the picture, but missing the "from left to right" part. Given a collection of such labeled photos, can a model learn to recognize individual faces? Similarly, consider a shopping cart and a printout from the register. Given a collection of unordered shopping carts together with matching receipts, can a model learn to recognize individual shopping items? These are example of a weakly-supervised learning where the goal is to learn to classify previously unseen feature vectors, but a training sample is a bag of feature vectors accompanied by a bag of correct labels, instead of a feature-vector and a correct label. We are not told which class belongs to which sample, which prevents us from directly using the standard cross-entropy loss.

More formally, consider a $d$-class classification problem, and a model $F(x_j; \beta)$ that for sample $x_j$ returns a $d$-dimensional vector of class probabilities, $p_j$, with $p_j^c$ denoting the predicted conditional probability of class $c$ given feature vector $x_j$. Let $y_j$ denote a $d$-dimensional, one-hot representation of the true class label of sample $x_j$, with $y_j^c = 1$ if sample $j$ is of class $c$, and zero otherwise. In weakly supervised learning involving bags of size $b$, we are given a tuple of $b$ feature vectors, $X = (x_j)_{j=1}^b$, and a tuple of permuted labels $Y = (y_{\sigma(i)})_{i=1}^b$ as one-hot-vectors, for some permutation $\sigma$; we will refer to the $j$-th element of the tuple $Y$ as $Y_j$. The permutation $\sigma$ is unknown, thus using a loss $\ell(p_j, Y_j) = \ell(p_j, y_{\sigma(i)})$ to compare predicted distribution over classes for sample $j$ with one-hot representation of $j$-th element in the randomly ordered set of true classes $Y_j$ makes no sense, since most likely $i \neq j$; $Y_j = y_{\sigma(i)}$ is the class for some other sample $i$, not for sample $j$.

While the permutation is unknown, with repeated presentation of bags of samples and bags of corresponding labels, we do have some information connecting the feature vector to classes. Intuitively, we can try to match model's outputs for feature vectors in the bag to the class labels using the information in the probability distribution $p_j$ over classes provided by the model for each feature vector $x_j$. That is, we can aim to find permutation $\hat{\sigma}$ optimal in the average loss sense $\min_{\hat{\sigma}} \sum_{j=1}^b \ell(p_j, \hat{\sigma}(Y)_j)$. If the class conditional probabilities $p_j$ resulting from the model perfectly match the one-hot vectors, the optimal $\hat{\sigma}$ will be the inverse of the permutation $\sigma$, that is, $\hat{\sigma}(Y)_j = y_j$.

---

**Algorithm 2** Loss based on bipartite matching for weakly-supervised image classification

---

**Input:** $X = (x_j)_{j=1}^b$ – bag of $b$ input images; $Y = (Y_k)_{k=1}^b$ – a set of $b$ sample classes to match, in one-hot representation, in arbitrary order; $\beta$ – ResNet18 network weights.
**Output:** Loss (optimal matching cost) and its generalized gradient, $L(\beta), \partial L(\beta)$
1: **procedure** MATCHBAG($X, Y, \beta$)
2:      forward pass, class probabilities $p_j = \mathrm{softmax}(\mathrm{ResNet18}(x_j; \beta))$ for $j = 1, ..., b$
3:      forward pass, cross-entropy for all image-label pairs $C_{jk} = \langle \log p_j, Y_k \rangle$ for $j, k = 1, ..., b$
4:      optimal matching cost and matching matrix: $z^*, M^* = \mathrm{OptMatching}(C)$,
             i.e., $M^* = \arg\min_M \langle C, M \rangle_F$, $z^* = \langle C, M^* \rangle_F$
5:      final loss: cost of optimal matching $L(\beta) = z^*$
6:      backward pass through bipartite matching $\partial z^*(C) = M^*$
7:      backward pass through cross-entropy, softmax and ResNet18: $\partial L(\beta) = M^* \frac{\partial C}{\partial \beta}$
8:      **return** $L(\beta), \partial L(\beta)$
9: **end procedure**

---

A $b$-element permutation can be represented by a $b \times b$ permutation matrix $M$. To find $M$, we define a $b \times b$ matrix $C$ with $C_{jk} = \ell(p_j, Y_k)$, where $\ell$ represents cross-entropy loss $\ell(p, y) = -\langle \log p, y \rangle$, with the logarithm applied element-wise. The elements $C_{jk}$ correspond to edge weight in a bipartite graph with the feature vectors $x$ processed by the neural network on one side, and labels $y$ on the other side. We use a combinatorial solver, for example the Hungarian method with computational complexity $O(b^3)$, to find the the permutation matrix $M^* = \arg\min_M \langle C, M \rangle_F$ minimizing the Frobenius inner product of $C$ and $M$. The procedure is outlined in Algorithm 2.

To test the approach, we used the CIFAR100 benchmark image dataset. As a baseline, we trained 5 independent fully supervised models with ResNet18 architecture (Zagoruyko & Komodakis, 2016) (see Supplementary Material for details), that is, models where each image is a separate sample with its true class available for loss calculation. To evaluate the ability of our method to provide gradients of a combinatorial loss defined by weighted matching, during training we explored image bags of samples consisting of $b$=4, 8, 12, 16, 24, or 32 images, and including correct but shuffled image labels. We trained 5 independent models for each bag size with the loss and its gradient provided using Algorithm 2. To avoid situations where the combinatorial loss is superficially aided by bags with mostly one class, we ignored any bag that has less than 75% of different classes, that is, for bag of size 8, we only consider bags that consist of at least 6 different classes. During testing, same as in the baseline model experiments, each image had the matching label available for test error calculations. For comparison, we trained a model with the same setup of image bags using cvxpylayers (Agrawal et al., 2019), a recently proposed methods for differentiable layers defined by conic programs. In contrast to our approach, which uses a combinatorial algorithm and relies on the LP formulation of the weighted bipartite matching only conceptually, for the definition of gradients, cvxpylayers solve the linear program in order to obtain gradients. We also trained the same model using a recently proposed approach to approximate gradients of the optimal solution vector, not the optimal objective value, of a combinatorial problem (Vlastelica Pogančić et al., 2020); we used the same combinatorial solver as in the experiments with our method.

Test error for CIFAR100 of the training set reshuffled into bags after each epoch (Fig. 1, left) shows that for bag sizes up to twelve elements, weak supervision through weighted bipartite graph matching is almost as effective as supervised learning with true label available for each individual image, that is, bag of size one. Training using the bipartite matching loss was implemented in three different ways: through interpolated combinatorial gradients proposed in (Vlastelica Pogančić et al., 2020), through differentiable LP approach (cvxpylayers), and through the proposed approach for obtaining gradients of the objective value. All three approaches lead to very similar error rates (Fig. 1, left), indicating these three ways of obtaining gradients provide similar training signal to the network. The two methods that use combinatorial solvers are much more efficient than LP solver-based cvxpylayers (Fig. 1, right). The performance of the LP-based method decreases for very small bag sizes, where each epoch has large number of individual problems to solve, as well as for large bag sizes, where each problem to be solved involves more computation. Among the two methods using the same combinatorial solver, our proposed method is twice as fast as the interpolation method of (Vlastelica Pogančić et al., 2020), which requires solving a combinatorial problem not only in the

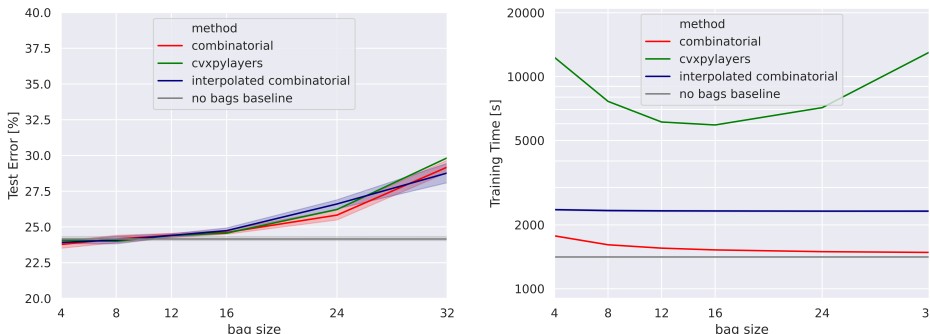

Figure 1: Test set error (left) and total training time (right) for increasing bag sizes for classifiers trained using the proposed bipartite matching loss with gradients calculated using the proposed approach and, for comparison, using cvxpylayers (Agrawal et al., 2019) and using an interpolation approach for obtaining gradients of the solution vector of combinatorial problems (Vlastelica Pogančić et al., 2020). A supervised model with true label available for each individual sample, which corresponds to bag of size one, is used as a baseline lower bound on the error that the bag-trained models should attempt to match. Mean, and the 95% confidence interval of the mean, are shown.

forward pass, but also in the backwards pass in order to obtain gradients of the solution vector. These results show that the generalized gradient over combinatorial optimization is effective in providing training signal to train a large neural network, and can do it much faster than the state-of-the-art alternative approaches.

## 3.2 Differentiating over Global Sequence Alignment for Sentence-level Loss in Sequence-to-Sequence Models

Another use case where a combinatorial loss is advantageous occurs in to sequence-to-sequence natural language models. We used a standard encoder-decoder architecture for the model (see Supplementary Material for details). The encoder takes the source sequence on input and prepares a context vector capturing the source sequence. The decoder is a recurrent network that outputs the predicted sequence one token at a time, based on the context vector and the output of the previous step. The output of the decoder at a step $t$ is a vector of probabilities $p_t$ over the set of all possible output tokens.

Existing encoder-decoder models use cross-entropy loss to compare predicted probabilities $p_t$ to the target word at position $t$, encoded as one-hot vector $y_t$. Instead of a sequence-level optimization, position-specific cross entropy loss results in an averaged token-level optimization. We hypothesize this has detrimental effect on the training process of differentiable sequence-to-sequence models that involve softmax or Gumbel-softmax (Jang et al., 2016) as the mechanism for feeding the output of the previous step of the decoder as input for the next step. For example, a recurrent model that learned to output almost all of the target sentence correctly but is still making the mistake of missing one word early in the sentence will have very high loss at all the words following the missing word – correcting the mistake should involve keeping most of the model and focusing on the missing word, but with position-specific loss, all the outputs are considered wrong and in need of correction.

Gaps or spurious words in the output sequence can be treated naturally if we consider global sequence alignment (GSA) as the loss. Global sequence alignment (Needleman & Wunsch, 1970) is a combinatorial problem in which two sequences are aligned by choosing, at each position, to either match a token from one sequence to a token from the other, or to introduce a gap in one or the other sequence; each choice has a cost (see Fig. 2). In sequence-to-sequence modeling, the cost of matching the decoder's output from position $i$ to the target sequence token as position $k$ will be given by $\langle -\log p_i, y_k \rangle$. The cost of a gap, that is, of a horizontal or a vertical move in Fig. 2, is specified in a way that promotes closing of the gap; we use the cost of diagonal move from that position as the cost of the gap, multiplied by a scalar $\gamma > 1$ to prioritize closing the gaps over improving the matchings. In our experiments, we used $\gamma = 1.5$. The GSA problem can stated as a linear program

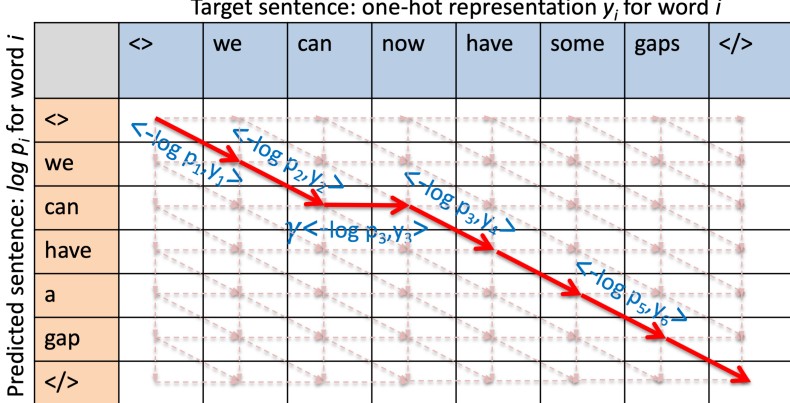

Figure 2: A directed acyclic graph (DAG) corresponding to the global sequence alignment between the target sequence and the sequence predicted by the RNN model. Each node, except the end of sequence indicator $</>$, has out-degree of three: a diagonal edge corresponding to a match between the predicted and the target sequence, a horizontal edge corresponding to a gap in the predicted sequence, and a vertical edge corresponding to a gap in the target sequence. Optimal sequence alignment is depicted in red, with the weights – the alignment costs – of the selected edges in blue.

with $p$ variables and $m+1$ constraints, with the costs of the moves forming the right-hand side of the constraints. Thus, by Theorem 1, the generalized gradient of the minimum global sequence alignment with respect to matching and gap costs is efficiently available.

In experiments involving global sequence alignment in sequence-to-sequence models, we used an encoder-decoder sequence-to-sequence architecture with bidirectional forward-backward RNN encoder and an attention-based RNN decoder (Luong et al., 2015), as implemented in PyTorch-Texar (Hu et al., 2018). While this architecture is no longer the top performer in terms of ROUGE metric – currently, large pre-trained self-attention models are the state-of-the-art – it is much more efficient in training, allowing for experimenting with different loss functions. During inference, we used beam search. During training, to have a differentiable decoder, we use two alternative approaches. First, we feed the probabilities resulting from the softmax layer applied to the outputs of the RNN directly as the recursive inputs to the RNN. Second, inputs to the RNN are provided by the straight-through Gumbel-softmax distribution (Jang et al., 2016) based on the outputs of the RNN, which is an approximation of the categorical distribution from which one-hot, single-token outputs are sampled. In both cases, as a baseline for comparisons with the GSA-based loss, we use word-level maximum likelihood, that is, cross-entropy between the probability vector on output of the softmax layer of the RNN and the desired target word at that position. In evaluating the combinatorial GSA loss, we used text summarization task involving the GIGAWORD dataset (Graff & Cieri, 2003) as an example of a sequence-to-sequence problem. We used test set ROUGE 1, 2, and L scores (Lin, 2004) as the measure of quality of the summarizations.

The results in Table 1 show that the GSA-based loss leads to improved text summarization results in all three ROUGE metrics compared to position-specific cross-entropy maximum likelihood training, both for the softmax and the Gumbel-softmax approach for providing the recursive input to the RNN in a differentiable way. The increase in accuracy comes at the cost of doubling the training time when our method is used to provide gradients of the optimal alignment score. A similar increased accuracy can be observed when the interpolation approach (Vlastelica Pogančić et al., 2020) for gradients of optimal alignment path is used instead, but the interpolation method further increases the training time, by a factor of two compared to our method. The proposed combinatorial approach is much more accurate and efficient than the recently proposed cvxpylayers method. The running time for the cvxpylayers approach is orders of magnitude slower. The cvxpylayers solver managed to reduce the training loss for several initial epochs, after which solver errors start to occur and the learning process diverges. In order to confirm this behavior, we performed 3 additional runs of the cvxpylayers-based training for the softmax model. In all cases, the loss dropped from the initial value in the 90-95 range to above 50, after which it increased to 500 or more. For comparison, the proposed combinatorial loss approach and the standard cross-entropy approach reach loss in the 30-32 range by epoch 10.

Table 1: Results for the GIGAWORD text summarization task using ROUGE-1, ROUGE-2, and ROUGE-L metrics. For the position-specific cross-entropy loss (MLE), for the interpolated combinatorial gradient (GSA-I) (Vlastelica Pogančić et al., 2020) applied to global sequence alignment, and for our combinatorial method (GSA-L), results are given as mean(std.dev.) over five independent runs with different random seed. For the method involving cvxpylayers (GSA-C) (Agrawal et al., 2019) applied to GSA, we only performed one run. We report test set values for the epoch that minimizes the total ROUGE score on a separate validation set. Time is per one epoch.

| Loss Type | ROUGE-Total | ROUGE-1 | ROUGE-2 | ROUGE-L | Epoch | Time |
|---|---|---|---|---|---|---|
| Softmax | | | | | | |
| MLE | 72.80(0.38) | 32.45(0.15) | 11.95(0.22) | 28.39(0.20) | 18.4(1.5) | 8 min |
| GSA - C | 32.18 | 17.04 | 2.49 | 12.65 | 3 | 9 hr |
| GSA - I | 75.87(0.82) | 33.94(0.31) | 12.03(0.35) | 29.90(0.32) | 13.4(3.4) | 32 min |
| GSA - L | 76.36(0.60) | 34.05(0.21) | 12.31(0.20) | 29.99(0.24) | 15.4(2.5) | 17 min |
| Gumbel-softmax | | | | | | |
| MLE | 67.50(0.20) | 31.25(0.18) | 9.72(0.26) | 26.52(0.08) | 18.0(2.8) | 9 min |
| GSA - I | 73.36(0.33) | 33.44(0.16) | 10.90(0.05) | 29.01(0.14) | 14.8(2.3) | 32 min |
| GSA - L | 72.62(0.51) | 33.25(0.15) | 10.60(0.22) | 28.77(0.17) | 14.0(1.9) | 17 min |

## 4 RELATED WORK

Recently, (Tschiatschek et al., 2018) proposed an approximate solver for submodular function maximization that uses differentiable elements and allows for differentiating through the solver. Differentiable solvers are also considered in (Mensch & Blondel, 2018), where dynamic programming solver is re-implemented with the maximum operation replaced by smoothed max. Similar approach is used in differentiable dynamic time warping (Chang et al., 2019). Several authors used a differential approximation to linear program solutions instead of introducing differentiable operations into combinatorial algorithms. WGAN-TS (Liu et al., 2018) solves an LP to obtain the exact empirical Wasserstein distance. Then, to circumvent lack of differentiability of linear programs, WGAN-TS proceeds by training a neural network to approximate the LP solution in order to obtain gradients. In seq2seq-OT (Chen et al., 2019), an approximation is used to model optimal transport between word embeddings serving as a regularizer in training sequence-to-sequence models. These approximation approaches are limited to specific problems and preclude using off-the-shelf combinatorial solvers.

Recently, an approach that relies on interpolation to obtain gradients of the optimal solution vector – not optimal objective value as in our method – produced by combinatorial solvers has been proposed (Vlastelica Pogančić et al., 2020; Rolínek et al., 2020). Similar to our approach, it allows for using off-the-shelf, black-box implementations of combinatorial algorithms. However, unlike our approach, it requires two executions of the solver, one in the forward phase, and a second execution for a slightly perturbed problem for the backward phase. As can be seen in our experiments, this results in doubling the performance overhead compared to our approach.

An alternative approach is to use mathematical programming solvers in gradient-trained neural networks. OptNet (Amos & Kolter, 2017) provides differentiable quadratic programming layers, and an efficient GPU-based batch solver, qpth. Cvxpylayers (Agrawal et al., 2019) generalizes this approach to a broad class of convex optimization problems expressed as cone programs, which include QP and LP as special cases, using conic solver based on ADMM, providing a general-purpose package based on the easy-to-use interface of cvxpy, with speed comparable to qpth for QP problems. Other authors (Wilder et al., 2019; Ferber et al., 2019) focus on LP problems, regularize them by adding the quadratic term, and use a QP solver as in OptNet to obtain the optimal solution vector and its gradient. Quadratic smoothing is also used in (Djolonga & Krause, 2017) in submodular set function minimization. While these methods can handle broader class of problems than our method, the reliance on quadratic or linear programming solvers translates to increased solving time. In the approach proposed here, linear programming is used only as a theoretical tool that allows for defining a mapping from the solution to a combinatorial problem to the gradient of its objective value. The solution is obtained by a single run of a combinatorial algorithm, which, as our experiments confirm, is faster than using mathematical programming and not affected by numerical instability and convergence problems.

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
