# OpenReview forum: "Differentiable Combinatorial Losses through Generalized Gradients of Linear Programs"
_ICLR.cc/2021/Conference — Reject_

### Official Review · AnonReviewer4 · 2020-10-27
**Differentiating over the objective of a linear (integer) program is not an interesting problem**

**Rating:** 3
**Confidence:** 5

**Review:**

The value of the optimal objective as a function of the cost vector $c$ can be written as $z^*(c) = c^T u^*(c)$ where the optimal solution $u^*$ also depends on $c$. The function $u^*(c)$ is piecewise constant -- there are finitely (resp. countably) many feasible solutions; candidates for $u^*$ -- and so the function $z^*(c)$ is a piecewise linear function of $c$, with gradient $u^*(c)$, wherever it exists (otherwise there is analogous subgradient). Obviously, all it takes for computing $u^*(c)$ is solving -- anyhow -- the combinatorial problem. This is all trivial and well-known, yet the authors do precisely that.

Can it be saved by proposing gradients also of w.r.t. constraints? No. These results are (slightly) less trivial but -- as authors admit -- are known since 1975. Moreover, the gradient with respect to $c$ is the only one used in experiments, as far as I understand.

Is there independent value in Theorem 1? I do not see it. It seems to be a bulky wrapper around the classical result. It only introduces some sort of transition from a vector specifying a combinatorial problem to a collection of vectors/matrices specifying an integer program. Also, the central concept of generalized gradient merely provides a formal framework to talk about non-unique gradients at boundary regions -- similarly to subgradient, subdifferential -- for the method itself, it has no specific relevance.

The claims of better performance compared to cvxpy are also absolutely non-surprising -- cvxpy currently uses a slightly suboptimal -- and a very expensive -- solver for linear programs. That is all.

---

> ### Author Response · Authors · 2020-11-25
> **Reply to reviewer 4**
>
> Thank you for a detailed critique; we have performed additional experiments to address your comment about the performance comparisons.
>
> To show performance benefits beyond the current implementations of differentiable convex programming layers, we have added the recently proposed approach for gradients of the solution vector of a combinatorial solver [Vlastelica Pogančić‬ et al., 2020] as another baseline. As expected, the proposed approach for differentiating the objective value is twice as fast as this baseline: it only solves the combinatorial problem once (in the forward pass), while the baseline method also solves a modified instance of the combinatorial problem in the backward pass, which they need to have the gradient of the optimal solution vector.
>
> These results confirm that narrowing the focus from the more general problem of solution vector gradient that has received attention recently to the objective value gradient and to existing mathematical techniques for obtaining it results in performance gains, even when in both cases combinatorial, non-LP-based solvers are being used.

---

### Official Review · AnonReviewer3 · 2020-10-27
**Well presented generic method for efficiently using combinatorial LP layers - albeit not completely novel**

**Rating:** 7
**Confidence:** 4

**Review:**

Summary
-------

The authors propose a simple method to optimize objective values defined as the
optimal value of a combinatorial integer linear program, whose parameter depends on the
output of a certain model.

For this, they note that generalized gradient of such objective values are
efficiently computed using the primal and dual solution of the ILP itself. In
particular, the ILP can be solved using specialized and efficient solver,
instead of solving a generic LP, as proposed in concurrent work.

The authors propose two example applications, that are described precisely, and validated against generic LP solving approaches. Using combinatorial specialized solvers outperform generic LP solving approaches in term of computation, and in term of validation metrics, as generic LP solving is hindered by errors which makes the learning process diverge.

Review
------

The paper is well written and well organized. The theoretical aspects are well documented, and the examples are introduced precisely and pedagogically.

The method itself is interesting as it ensures that the generalized gradient of
many ILP problems are computable efficiently. Theorem 1 states those guarantees, and many examples are discussed.

On the other hand, the novelty of the method may be a little overstated. In
particular, it is known that using a generic LP solver is oftentimes not the
most efficient way of computing the gradient, and that specialized combinatorial
solver should be used.

For the problem of GSA, which corresponds to using a dynamic time warping loss,
solving the small LP is done using dynamic programming. Using a DTW loss on top
of a deep neural network has already been studied (see e.g. the cited Mensch and
Blondel paper, where the authors solve the LP using DP). Using a generic LP
solver such as the one in cvxpy is a little naive in that case, and it not
surprising that it performs poorly.

In this example, we only require the gradient with respect to the cost (P is
"primal-pdfiff-efficient"). Arguably, this manuscript also enable us to
backpropagate through parametrized constraints, using the formula proposed in
Theorem 1, which is little known in this community.

---

> ### Author Response · Authors · 2020-11-25
> **Reply to reviewer 3**
>
> Thank you for your comments and suggestions. We address them below:
>
> >> On the other hand, the novelty of the method may be a little overstated. In particular, it is known that using a generic LP solver is oftentimes not the most efficient way of computing the gradient, and that specialized combinatorial solver should be used. (...) For the problem of GSA, which corresponds to using a dynamic time warping loss, solving the small LP is done using dynamic programming. Using a DTW loss on top of a deep neural network has already been studied (see e.g. the cited Mensch and Blondel paper, where the authors solve the LP using DP). Using a generic LP solver such as the one in cvxpy is a little naive in that case, and it not surprising that it performs poorly.
>
> The method by Mench and Blondel relies on re-implementing dynamic programming with the maximum operation replaced by its smoothed version. The method we propose works with existing combinatorial or LP solvers. To provide a comparison beyond LP solvers implemented in cvxpylayers, in the updated manuscript we added experiments with a recently proposed method for differentiating the combinatorial problem's solution vector using a combinatorial solver [Vlastelica Pogančić‬ et al., 2020]. To find the gradient of the solution vector, that approach requires solving two instances of the combinatorial problem, one in the forward phase, one in the backwards phase. Our method, by focusing on the gradient of the objective value only, which is what we need in the analyzed use cases, requires only one call to the combinatorial solver; experiments conclude that it is twice as fast.

---

### Official Review · AnonReviewer1 · 2020-10-28
**Novel approach, comparisons don't compare to other combinatorial optimization differentiation methods**

**Rating:** 6
**Confidence:** 3

**Review:**

This paper shows how to differentiate through combinatorial
losses by differentiating through the ideal formulation LP.
Understanding how to differentiate through combinatorial
optimization so that it can be used as part of the model or
loss is important as it captures many natural operations.
I am giving this a weak accept as it is a novel approach
for differentiation that the community can build on,
but the positioning and relation to prior work and
empirical comparisons could be stronger (more details below).

# Strengths
To the best of my knowledge this is a novel approach that
makes the elegant and natural connections going from
a combinatorial problem to an ILP to an LP
to differentiating through the LP using known methods.

# Weaknesses
The biggest weakness is the lack of a comparison with related
approaches for differentiating through combinatorial losses,
such as some of the approaches discussed in the introduction
as [Pogancic 2020] that consider similar problems.
The experimental settings considered in this paper compare to
baselines that *don't* differentiate through the combinatorial
aspect of the problem. While this is a great step of validating
the power of these approaches, I think that it would be significantly
more convincing to empirically compare to approaches that
differentiate through the combinatorial losses.

I also think it's important to discuss the comparisons to the
related approaches for differentiating through parameterized
combinatorial optimization. Are the approaches using the same
definition of a derivative? [Pogancic 2020] discusses an issue
with the real derivative through combinatorial optimization being
uninformative or near-zero everywhere, is this also an issue in
the setting here?
Can this approach be seen as an approximation or surrogate to
the derivative of the combinatorial problem as the other approaches?

If I understand correctly, this approach requires a known mapping from
the combinatorial problem to the ILP, and from the ILP to the LP,
which could make it more involved to apply than some of the related
methods that don't require knowing this information.

# Other questions and comments
How should the gradients of the continuous baselines (with CVXPY)
compare to the method being proposed in the experiments?
If they're using the ideal formulation LP, should they be the
same in theory (as Figure 1 validates), but in practice due
to solver errors, gives suboptimal directions (as Table 1 shows)?

The last paragraph of the introduction presents a form of the
criterion with a loss and the combinatorial objective value
with notation that's not used later on in the paper.

Page 2, second paragraph: The last sentence on differentiable
continuous LPs/QPs seems separate from the rest of paragraph
on combinatorial solvers.

---

> ### Author Response · Authors · 2020-11-25
> **Reply to reviewer 1**
>
> Thank you for critique and suggestions for improvements. We address them below:
>
> >> The biggest weakness is the lack of a comparison with related approaches for differentiating through combinatorial losses, such as some of the approaches discussed in the introduction as [Pogancic 2020] that consider similar problems. (...) it would be significantly more convincing to empirically compare to approaches that differentiate through the combinatorial losses.
>
> We have expanded the manuscript to include experiments with the approach proposed in [Pogancic 2020]. As expected, their approach is twice as slow, due to the fact that it requires two calls to the combinatorial solver instead of one in our approach. The need for the second call in [Pogancic 2020] arises from the difference in the goals of the two approaches: ours is focused on the gradient of the objective value directly, while theirs is focused on the gradient of the solution vector, from which the objective value can be obtained if needed via a differentiable operation.
>
> >> Are the approaches using the same definition of a derivative?  [Pogancic 2020] discusses an issue with the real derivative through combinatorial optimization being uninformative or near-zero everywhere, is this also an issue in the setting here? Can this approach be seen as an approximation or surrogate to the derivative of the combinatorial problem as the other approaches?
>
> The gradient of the solution vector is null everywhere, and requires interpolation as in [Pogancic 2020]. On the other hand, the gradient of the objective value is not constant, and we do not need any interpolation/approximation.
>
> >> How should the gradients of the continuous baselines (with CVXPY) compare to the method being proposed in the experiments? If they're using the ideal formulation LP, should they be the same in theory (as Figure 1 validates), but in practice due to solver errors, gives suboptimal directions (as Table 1 shows)?
>
> Indeed, the gradient resulting from our approach and the gradient resulting from the use of cvxpylayers should in principle be the same.
>
> >> If I understand correctly, this approach requires a known mapping from the combinatorial problem to the ILP, and from the ILP to the LP, which could make it more involved to apply than some of the related methods that don't require knowing this information.
>
> The mapping of the combinatorial problem to a corresponding LP is indeed needed, conceptually, to specify how the parameters defining the combinatorial problem relate to the vectors/matrices specifying the LP and, for the gradients, how the combinatorial solution vector relates to the primal/dual solution vectors. That does not mean, however, the detailed LP for a specific instance needs to be spelled out.
>
> >> The last paragraph of the introduction presents a form of the criterion with a loss and the combinatorial objective value with notation that's not used later on in the paper. (...) Page 2, second paragraph: The last sentence on differentiable continuous LPs/QPs seems separate from the rest of paragraph on combinatorial solvers.
>
> Thank you for pointing it out - we have edited the manuscript to remove these inconsistencies.

---

### Official Review · AnonReviewer2 · 2020-11-01
**review for "Differentiable Combinatorial Losses through Generalized Gradients of Linear Programs"**

**Rating:** 8
**Confidence:** 3

**Review:**

The authors present a technique to integrate combinatorial optimization sub-problems into a gradient descent based application. The approach they describe relies only on differentiation of the value of the combinatorial program (instead of the solution vector), and can be done with relatively low overhead (compared to techniques that involve modifying combinatorial algorithms to differentiable elements, or the use of differentiable linear/quadratic programming layers)

They motivate and show the advantages of their approach using two natural and useful examples. The experimental results show promise, and the paper is well written, and motivated.

---

> ### Author Response · Authors · 2020-11-25
> **Reply to reviewer 2**
>
> Thank you for your comments.
>
> We have added a comparison with a recently proposed method for differentiating the solution vector using combinatorial solvers, to further illustrate the performance benefits resulting from the proposed approach.

---

### Official Review · AnonReviewer5 · 2020-11-06
**Interesting general method, but needs far more details about the resulting algorithm for concrete use-cases.**

**Rating:** 5
**Confidence:** 4

**Review:**

=quality=Proposed method is significantly more practical, both in terms of ease of implementation and speed, than prior related work for minimizing combinatorial losses. The paper's exposition needs to be improved considerably, though (see below).

=originality=The paper draws on advanced concepts from combinatorial optimization that may be unfamiliar to many ICLR readers, but that have the potential for large impact.

= significance=The paper's proposed method is practical for very common ML setups in NLP and computer vision that are used day-to-day.

= Pros = Proposed method is interesting, practical, and relatively easy to implement.

= Cons = Paper writing omits many key details necessary to use the method in practice. Experiments build the method on top of out-
dated models and do not demonstrate that the method could be used with modern models (e.g. attention-based decoders).

==Comments==I appreciate that you provide a very general recipe for constructing the differentiable combinatorial layers. However, the paper provides far too few details for the particular problems (bipartite matching and sequence alignment) that appear in the experiments. The supplementary material does not help. Your method is promising, and practitioners that do not have a background in combinatorial optimization may want to use it. The paper does not provide enough details to do so. I'd replace Algorithm 1 with a box specific to bipartite matching.

You need to provide far more detail/background on differentiable decoding for the secnod set of experiments. It was unclear what Softmax/ Gumbel-Softmax meant. Is Softmax not the same as MLE?

While the second set of experiments provides useful ablation analysis of the impact of your method, it builds on an out-dated model. You write "while this architecture is no longer the top performer in terms of ROUGE metric –currently, large pre-trained self-attention models are the state-of-the-art – it is much more efficient intraining, allowing for experimenting with different loss functions." Is the speed difference really that much? I'm surprised that that makes a difference in terms of which experiments are feasible.

Figure 1: why is cvxpy timing u-shaped?

---

> ### Author Response · Authors · 2020-11-25
> **Reply to reviewer 5**
>
> Thank you for your comments and suggestions for improvements. Below, we address each of them:
>
> >> ... practitioners that do not have a background in combinatorial optimization may want to use it. The paper does not provide enough details to do so. I'd replace Algorithm 1 with a box specific to bipartite matching.
>
> We have added Algorithm 2 with the outline of the procedure for bipartite matching example. In addition, in the Supplementary Material, we show the pytorch code for the Autograd class implementing differentiation of the optimal matching cost. We are preparing a github repository that will include the full code and experiments, not only for the bipartite matching but also for the language modeling experiments.
>
>
> >> You need to provide far more detail/background on differentiable decoding for the secnod set of experiments. It was unclear what Softmax/ Gumbel-Softmax meant. Is Softmax not the same as MLE?
>
> We edited the manuscript to clarify the difference between these two approaches for providing the input to the next iteration of an RNN in a differentiable way. While the softmax approach directly uses the distribution over the vocabulary from the softmax output layer of the RNN as the input to the next step in the RNN for next word decoding, Gumbel-softmax uses instead a differentiable sample from that softmax-based output distribution. In both cases, for a single predicted word, we use a standard MLE/softmax cross-entropy approach for assessing the prediction loss.
>
> >> You write "while this architecture is no longer the top performer in terms of ROUGE metric –currently, large pre-trained self-attention models are the state-of-the-art – it is much more efficient intraining, allowing for experimenting with different loss functions." Is the speed difference really that much?
>
> Large seq2seq models like GPT-2 that are pre-trained on the task of predicting the next word (i.e., the desired output is a shifted sequence) might benefit from using sentence-level pre-training via global sequence alignment and the combinatorial gradients, but our computational budget did not allow us to perform GPT-2 pre-training to test this hypothesis.
>
> >> Figure 1: why is cvxpy timing u-shaped?
>
> We expanded the manuscript to provide an explanation of this phenomenon. For a sample bag of size b = 4,...,32, in each epoch involving m=50,000 images there are m/b individual bags, that is, b-to-b bipartite matching problems to be solved. For all methods, we see an initial drop in total time as the number of individual problems drops when b increases, and the time to solve each of the problems increases only moderately. But for large values of b, the time to solve each combinatorial problems grows more steeply for cvxpylayers, overtaking any gains from reducing the number of individual problems. This is not the case for the combinatorial solver, where increase in time to solve one problem is offset by the reduction in the number of problems, and the total time stays stable in the analyzed b=4,...,32 range.

---

### Decision · Program_Chairs · 2021-01-07
**Final Decision**

**Decision:**

Reject

**Comment:**

This paper received high variance in the reviews.

I personally agree with AnonReviewer4 that the theoretical results presented in this paper are well-known results on the sensitivity analysis of linear programs. See for instance "Introduction to linear optimization" by Bertsimas and Tsitsiklis, Chapter 5.

More generally, these results are a special case of Danskin's theorem and the envelope theorem:
https://en.wikipedia.org/wiki/Danskin%27s_theorem
https://en.wikipedia.org/wiki/Envelope_theorem

Clarke's generalized gradients are just subgradients in the case of convex functions, which is the case here.

My recommentation to the authors if they want to publish their work is to focus on the applications and to stop claiming novelty on the theoretical side.